# EUS-Guided Gastroenterostomy in Malignant Gastric Outlet Obstruction: A Comparative Study between First- and Second-Line Approaches after Enteral Stent Placement

**DOI:** 10.3390/cancers14225516

**Published:** 2022-11-10

**Authors:** Enrique Perez-Cuadrado-Robles, Hadrien Alric, Ali Aidibi, Michiel Bronswijk, Giuseppe Vanella, Claire Gallois, Hedi Benosman, Emilia Ragot, Claire Rives-Lange, Gabriel Rahmi, Christophe Cellier

**Affiliations:** 1Department of Gastroenterology, Georges-Pompidou European Hospital, 75015 Paris, France; 2Faculty of Medicine, University of Paris-Cité, 75006 Paris, France; 3Department of Gastroenterology and Hepatology, University Hospital Gasthuisberg, University of Leuven, 3000 Leuven, Belgium; 4Department of Gastroenterology and Hepatology, Imelda Hospital Bonheiden, 2820 Bonheiden, Belgium; 5Pancreatobiliary Endoscopy and Endosonography Division, IRCCS San Raffaele Scientific Institute, 20132 Milan, Italy; 6Department of Oncology, Georges-Pompidou European Hospital, 75015 Paris, France; 7Department of Surgery, Georges-Pompidou European Hospital, 75015 Paris, France; 8Department of Nutrition, Georges-Pompidou European Hospital, 75015 Paris, France

**Keywords:** gastric outlet obstruction, pancreatic cancer, endoscopic ultrasound, LAMS

## Abstract

**Simple Summary:**

Endoscopic ultrasound-guided gastroenterostomy (EUS-GE) is increasingly used in the setting of malignant gastric outlet obstruction (GOO). The aim of the present observational study is to compare the outcomes of EUS-GE by using the freehand technique as a first- and second-line approach after enteral stenting (ES). The primary outcome was the clinical success, defined as a solid oral intake at 1 week after the procedure (GOO Score, GOOSS ≥ 2). The secondary outcomes were technical success and adverse event (AE) rates. Twenty-eight patients with (*n* = 13, 46.4%) and without (*n* = 15, 53.6%) a previous ES were included. The diet progression was quicker in patients with a previous ES (GOOSS at 48 h, 2 vs. 1, *p* = 0.023), but the GOOSS at 1 week (*p* = 0.299), albumin gain (*p* = 0.366), and BMI gain (0.257) were comparable in the two groups. The technical success was achieved in 25 cases (89.3%), with no differences between the two groups (92.3% vs. 86.7%). The AE rate was 7.1%. In conclusion, primary EUS-GE might require fewer procedures and less discontinuation of chemotherapy to achieve a comparable result.

**Abstract:**

Introduction: Endoscopic ultrasound-guided gastroenterostomy (EUS-GE) is increasingly used in the setting of malignant gastric outlet obstruction (GOO). However, little is known about the role of primary EUS-GE. The aim of the present study is to compare the outcomes of EUS-GE by using the freehand technique as a first- and second-line approach after enteral stenting (ES). Methods: This is an observational single-center study using a prospectively collected database. All consecutive patients who underwent an EUS-GE using the freehand technique due to malignant GOO were included. Patients with previous gastric surgery, a wire-guided EUS-GE technique, or those presenting without GOO were excluded. The primary outcome was the clinical success, defined as a solid oral intake at 1 week after the procedure (GOO Score, GOOSS ≥ 2). The secondary outcomes were technical success and adverse event (AE) rates. The impact on nutritional parameters was also assessed. Results: Forty-five patients underwent an EUS-GE for all indications. Finally, 28 patients (mean age: 63 ± 17.2 years, 57.1% male) with (*n* = 13, 46.4%) and without (*n* = 15, 53.6%) a previous ES were included. The technical success was achieved in 25 cases (89.3%), with no differences between the two groups (92.3% vs. 86.7%, *p* = 1). The median limb diameter and procedure time were 27 mm (range:15–48) and 37 min. Overall, clinical success was achieved in 22 cases (88%), with three failures due to AEs (*n* = 2) or peritoneal carcinomatosis (*n* = 1). The diet progression was quicker in patients with a previous ES (GOOSS at 48 h, 2 vs. 1, *p* = 0.023), but the GOOSS at 1 week (*p* = 0.299), albumin gain (*p* = 0.366), and BMI gain (0.257) were comparable in the two groups. The AE rate was 7.1%. Conclusions: EUS-GE achieves a high technical and clinical success in patients with GOO regardless of the presence of a previous ES. Patients with previous ES may have a quicker progression of their diet, but the GOOSS and nutritional status in the long term at 1 week or 1 month are comparable. Primary EUS-GE might require fewer procedures and less discontinuation of chemotherapy to achieve a comparable result.

## 1. Introduction

Endoscopic ultrasound (EUS)-guided therapies are gaining ground in the armamentarium of interventional endoscopy [1]. The creation of digestive anastomosis by using lumen apposing metal stents (LAMS) has greatly impacted the decision-making strategy in selected patients with altered anatomy (e.g., gastric bypass, afferent limb syndrome) [2,3], difficult biliary drainage [4], and gastric outlet obstruction (GOO) [5]. EUS-guided gastroenterostomy (EUS-GE) [6], in particular, has been proposed as an alternative in the treatment of malignant GOO and will probably be the mainstay in the management of this disease in the near future. Although prospective data in this setting are scarce, a systematic review and meta-analysis of 12 studies including 290 patients concluded high (>90%) technical and clinical success rates [7]. In addition, early data based on retrospective series suggest that EUS-GE has a lower complication rate compared to surgery [8,9], even in the presence of peritoneal carcinomatosis [10]. 

Compared to enteral stenting (ES), EUS-GE has theoretical advantages, such as longer patency, higher clinical success rates and similar safety while allowing a more advanced diet [11]. Indeed, ES increases the risk of secondary biliary obstruction, and the tumor progression can lead to recurrent GOO with a higher number of reinterventions. Finally, EUS-GE could be optimal in patients with long, multifocal, or distal stenosis, whereas ES may be less effective. These are probably some of the reasons why recent ESGE [12] and ACG guidelines [13] recommend EUS-GE as an alternative to ES or surgery for malignant GOO, when relatively longer life expectancies can be predicted. 

Several EUS-GE techniques, such as “freehand” LAMS placement [14], the wire-guided EUS-GE, and the EUS-guided double-balloon-occluded gastroenterostomy bypass (EPASS) [15], have been described, with initial data showing no influence of the technique on the outcomes and the risk of adverse events [16]. The wireless EUS-GE simplified technique (WEST) has been proposed as a safer technique without the need of a confirmatory puncture by a 19-gauge needle or guidewire cannulation that can increases costs and procedure duration and may lead to a false sense of security [17,18]. 

However, EUS-GE often remains a second-line therapy in daily practice, and long-term data are lacking. Moreover, although this technique will probably be considered as a standard of care in a near future [19], few case series reported the use of EUS-GE as a first-line approach. The aim of the present observational study is to compare the outcomes of EUS-GE using the WEST technique as a first- and second-line approach following ES. 

## 2. Methods

### 2.1. Patients

This is an observational retrospective single-center study using a prospectively collected database. All consecutive adult patients who underwent a EUS-GE using the WEST technique due to malignant GOO at Georges-Pompidou European Hospital, Paris, between January 2020 and September 2022 were included. Patients with previous gastric surgery, or those who underwent a wire-guided EUS-GE technique were excluded. Similarly, those patients who underwent a EUS-GE for other reasons other than the palliation of GOO (e.g., transgastric access to biliary limb in patients with post-surgical anatomy, drainage of the afferent limb syndrome) were excluded. The protocol was submitted to the Local Ethical Committee (CERUPHO) and was approved by the national data protection commission (CNIL) according to MR-004 reference methodology (no. 2212878 v 0).

Age, sex, and demographic variables were collected. Baseline characteristics such as underlying cancer primary site, metastatic status, presence of peritoneal carcinomatosis, length of hospital stay, and ascites were noted. Time elapsed between the GOO symptoms and the procedure was also considered. The Gastric Outlet Obstruction Scoring System (GOOSS) was evaluated 24 h before, 48 h after, and one week after the procedure as follows: 0 in case of no oral intake, 1 for only liquids, 2 for soft solids, and 3 for low residues or full diet. An additional evaluation of the GOOSS was considered at the end of follow-up. The need of a complementary enteral nutrition was assessed. 

### 2.2. The EUS-GE Procedure

A linear-array echoendoscope (GF-UE180, Olympus, Tokyo, Japan) was used to create the anastomosis with an electrocautery enhanced LAMS of 20 mm lumen diameter (20 × 10 mm Hot-Axios, Boston Scientific, Marlborough, MA, USA) by using Autocut Effect 4 (VIO 3, Erbe Elektromedizin GmbH, Tuebingen, Germany). 

First, a 3.7-mm working channel gastroscope was used to cannulate the duodenal stenosis using a 7 French tandem catheter allowing either contrast opacification or passing a 0035 in guidewire (Jagwire, Boston Scientific) in the first jejunal limb to understand the length of the stenosis. Thus, a 6 French single cannula was left in place at the angle of Treitz while the gastroscope was removed. This oroenteric catheter, connected to a water pump, allows one to irrigate the small bowel lumen and guide further EUS-guided therapy. Finally, an EUS-GE using the freehand (WEST) technique by direct perforation of the saline-filled small bowel target was performed (Figure 1). Preferably, the angle of Treitz was chosen for creating the anastomosis because this area is “more fixed” compared to distal jejunal limbs with a lower risk of “pushing” away the jejunal limb. EUS guidance for the whole placement of the stent (distal and proximal flanges) was performed. An anticholinergic agent was also used to reduce the motility and facilitate LAMS insertion. 

All procedures were performed under general anesthesia and CO_2_ insufflation. All patients received antibiotics periprocedurally. Radiological guidance was used in all cases. No LAMS dilation was performed during the baseline procedure. 

The EUS characteristics (diameter of the limb), the distance from the gastric wall to the small bowel wall (<1 cm, 1–2 cm, >2 cm) and the location of the gastroenteric anastomosis (D3, D4, angle of Treitz, first jejunal limb, distal jejunum) were considered. The presence of a previous uncovered duodenal stent and the length and location of the malignant stenosis were noted. Procedure time was also assessed from the insertion to endoscope removal. All procedures were performed by a single endoscopist (EPCR). 

### 2.3. Outcomes 

The primary outcome was the clinical success, defined as a solid oral intake at 1 week after the procedure (GOOSS ≥ 2). 

The secondary outcomes were technical success and AE rates. Technical success was defined as the successful placement of the LAMS with the creation of an anastomosis between the stomach and the duodenal or jejunal lumen below the malignant stricture. AEs were graded according to the AGREE classification [20]. 

A comparative analysis of these outcomes was carried out considering patients in two groups with and without a previous uncovered duodenal stent. Weight, BMI, and albumin levels were assessed 24 h before EUS-GE and 1 month later.

### 2.4. Follow-Up

A clinical follow-up was considered in all cases. Recurrent GOO, defined as the occurrence of GOO symptoms with a GOOSS < 2 in those patients with a previous clinical success, was considered. 

### 2.5. Statistical Analysis

Categorical variables were compared by using *χ*^2^ or Fisher exact tests. Non-normally distributed continuous variables were analyzed by Mann–Whitney U-test or McNemar test. Normal and non-normal variables were presented as mean (SD) and median (range). The impact on nutritional status and GOOSS score according to the presence of a previous duodenal stent was analyzed by a per-protocol analysis strategy considering only patients with previous EUS-GE technical success. A two-sided *p*-value < 0.05 was considered statistically significant. SPSS software v.24 was used (IBM, SPSS Inc., Chicago, IL, USA).

## 3. Results

### 3.1. Patients

Forty-five patients underwent an EUS-GE during the period of study. Seventeen were excluded because of afferent limb syndrome indication (*n* = 7), EUS-GE in GOO using the wire-guided technique (*n* = 5), EUS-GE for transitory access to the biliary limb (*n* = 3), and other EUS-GE in patients with previous gastric surgery (*n* = 2). Finally, 28 patients (mean age: 63 ± 17.2 years, 57.1% male) with (*n* = 13, 46.4%) and without (*n* = 15, 53.6%) previous ES were included. Most of malignant stenosis were located in the proximal duodenum (D1–D2, *n* = 16, 57.1%), followed by the distal duodenum (D3–D4, *n* = 7, 25%) and the angle of Treitz (*n* = 4, 14.3%). The two patients with ascites did not present with peritoneal effusion in the tract of the endoscopic anastomosis but only surrounding the liver parenchyma. Patient baseline characteristics are shown in Table 1. The median elapsed time between ES placement and EUS-GE for those patients with a previous duodenal stent was 18 days (range: 5 days–4.7 months).

### 3.2. Impact of the Procedure

Technical success was achieved in 25 cases (89.3%), with no differences between patients with or without a previous duodenal stent (92.3% vs. 86.7%, *p* = 1). Thus, three patients had a technical failure due to an infiltration of the gastric wall in a patient with gastric cancer (*n* = 1) and a long distance of the jejunal limb from the gastric wall (>1 cm) with peritoneal carcinomatosis (*n* = 2). 

The median limb diameter after using the water pump was 27 mm (range: 15–48). Notably, there were 13 cases (46.4%) with a <30 mm diameter despite the use of the water pump by the oroenteric catheter. Similarly, most of patients (*n* = 19, 67.9%) presented with a <1 cm distance between the gastric and small bowel walls; however, there were cases with ≥1<2 cm (*n* = 6, 21.4%) and >2 cm distances (*n* = 3, 10.7%). There were no statistically significant differences in the distances when a duodenal stent was present or not (*p* = 0.262). The anastomosis was performed in the first jejunal limb (*n* = 17, 60.7%), angle of Treitz (*n* = 8, 28.6%), or distal duodenum (*n* = 3, 10.7%). There was no association between the performance of the EUS-GE in the first jejunal limb compared to the angle of Treiz/duodenum and a history of ES (47.1% vs. 52.9%, *p* = 0.891) or the impact in clinical success (82.4% vs. 100%, *p* = 0.205). The median procedure time was 37 min (20–52), with comparable times regardless of previous ES (35 vs. 37 min, *p* = 0.511). 

The mean weights of patients before and after EUS-GE were 55.9 ± 11.1 kg and 56.1 ± 9.7 (*p* = 0.640). Overall, clinical success was achieved in 22 cases (78.6%). However, considering only the 25 patients with technical success, the clinical success was achieved in all but three patients (88%) due to intraoperative AEs (*n* = 2) or peritoneal carcinomatosis with a distal jejunal stenosis not detected by a previous CT scan (*n* = 1). There was no statistically significant association between peritoneal carcinomatosis and clinical success (*p* = 0.622). 

The impact on the nutritional status of the technique based on the presence of a previous duodenal stent in patients with technical success is shown in Table 2. Notably, the diet progression was quicker in patients with a previous ES, as the median GOOSS at 48 h after EUS-GE was higher in this group (2 vs. 1, *p* = 0.023), but all other parameters were comparable. A complementary artificial nutrition was able to be withdrawn in 5 of 12 cases, and only 7 patients had enteral (*n* = 6) or parenteral nutrition (*n* = 1) at the end of the follow-up, with no differences between the two groups (*p* = 0.753). 

### 3.3. Safety of EUS-GE

Overall, the AE rate was 7.1% as there were two patients with severe and fatal AEs. Although both of them had a history of previous ES, there was no statistically significant difference between both groups (0 vs. 13.33%, *p* = 0.225), and the AEs were not related to the duodenal stent. A 59 year-old man who underwent a EUS-GE successfully performed and confirmed in CT scan, presented with a colonic perforation and a gastro-colonic anastomosis 12 days later, probably due to the perforation and crossing of the left colon while performing the gastrojejunostomy, resulting in a delayed migration of the stent in the colon lumen. This complication led to the death of the patient (AGREE IV). Another 76 year-old patient with pancreatic cancer who underwent a technically successful EUS-GE in the first jejunal limb presented with a delayed colonic perforation of the transverse colon suspected during the endoscopic procedure, leading to a surgical intervention a few hours later (AGREE IIIb). No intraoperative or procedure-related delayed bleeding was retained. 

### 3.4. Follow-Up

The median follow-up of the whole cohort was 4 months (range: 1–22). Recurrent GOO was detected in 4 patients with previous clinical success (18.2%) due to either downstream peritoneal carcinomatosis (*n* = 3) or tumoral progression with secondary occlusion of the LAMS (*n* = 1). In this last case, a duodenal stent was placed (“stent in stent” technique) with further clinical improvement and palliation of the GOO. Thus, procedure-related recurrent GOO was only present in one case (4.5%). In addition, there were three patients with a functional EUS-GE (13.6%) that decreased the GOOSS to 1 without symptoms of GOO at the end of the follow-up, probably due to the advanced oncological status. 

## 4. Discussion

In this observational study, we presented 28 patients with malignant GOO who underwent EUS-GE using a standardized WEST technique, achieving technical and clinical success rates of 89.3% and 88%. Approximately half of the patients had undergone previous ES. The evolution of the nutritional outcomes (BMI, albumin, weight) was comparable in patients with and without previous ES. However, diet progression at 48 h assessed by the GOOSS was quicker in the first group (2 vs. 1, *p* = 0.023). No differences were found in the long term GOOSS at 1 week, at the end of the follow-up or the GOOSS gain. The AE rate was 7.1% and mortality was 3.6%. 

EUS-GE may overcome some advantages compared to ES, while avoiding gastroparesis and morbidity of surgical anastomosis [18]. However, most data to date include patients with previous ES and recurrent GOO, who may present with worse outcomes of the technique compared to “naive” patients. In this sense, performing EUS-GE as first line in selected cases could avoid unnecessary invasive procedures under general anesthesia, prevent gastric decompensation, decrease the AEs related to recurrent GOO after ES placement [11], and shorten the length of hospitalization and impact on nutritional status at earlier stages, ensuring improved quality of life and undelayed oncological therapy. In our center, the decision-making strategy to decide whether to perform ES or EUS-GE at first line is decided in a multidisciplinary setting, also considering the willingness of the patient. For those cases presenting with long or multifocal strictures, previous biliary stenting or previous ES, an EUS-GE is preferred. In addition, in our series, the 25% of patients had nonextensive peritoneal carcinomatosis, but there was a unique stenosis with no further strictures in the distal small bowel in all cases. The procedure was performed by using a pre-standardized protocol avoiding the wire-guided technique, as these excessive procedural steps and device exchanges may increase the risk of AEs [21]. Indeed, standardizing the procedure is associated with a significant increase in technical success and a decrease in AEs irrespective of prior total experiences [22]. We performed no intraoperative LAMS dilation, and no further procedures were needed to dilate the stent in the follow-up. Finally, liquids were initiated 24 h after the procedure, followed by a progressive soft solid diet from 48 h after the procedure onwards. 

The baseline characteristics of patients with and without a previous ES were comparable in our study. Notably, the median elapsed time between the symptoms and the EUS-GE was short in both groups (4 and 6 days). A recent multicenter retrospective study including 97 patients comparing EUS-GE and ES concluded that EUS-GE seems to have improved patency outcomes for palliative treatment of malignant GOO [23]. In our series, although the baseline GOOSS was also comparable, the GOOSS improvement at 48 h was higher in patients with a duodenal stent. This difference could be explained by the “double route” of patients with a partially functional ES before EUS-GE and physician bias. There were three patients in the ES group (25%) with a baseline GOOSS score at 1, whereas all patients in the “naive” group had a baseline GOOSS at 0. However, the middle- and long-terms effectiveness of the EUS-GE at one week or at the end of follow-up were comparable between both groups, with an overall clinical success of 88%. In addition, it is important to highlight that all clinical failures in our series were due to AEs leading to prolonged fasting (66%), or unmasking of a distal stenosis due to peritoneal carcinomatosis (33%). Thus, this technique is expected to be effective in patients who underwent a technically successful anastomosis with no major AEs or distal stenosis. 

The assessment of the impact of this technique in the long term is challenging, as there are many factors and conditions related to the oncological progression (e.g., pancreatic cancer-related pain, anorexia) and chemotherapy treatment that may have a major role in the diet progression and nutritional status regardless of the functionality of the EUS-GE. This is probably the main reason explaining why the 28% of patients needed a complementary artificial nutrition at the end of follow-up in our study. An increased risk of GOO recurrence with ES versus EUS-GE has been recently described [24]. Similarly, there was only one patient with an EUS-GE dysfunction in our series (invasion of the stent by peritoneal carcinomatosis, 4%). 

Concerning the safety of the procedure, recent meta-analysis on EUS-GE have reported periprocedural AEs in 12% of cases [7,25]. Most of the reported AEs were graded as mild or moderately severe, including stent maldeployment resulting in perforation with peritonitis, stent misplacement and stent migration or dislodgement [12], but fatal events have also been described. Endoscopic salvage therapy is successful in most cases [6], provided this is detected intraoperatively. In our study, to ensure a homogeneous population, we have only included patients who underwent EUS-GE using the WEST technique following a standardized protocol after a learning curve. Two major AEs (7.1%) were identified, both due to a colonic perforation during the gastro-jejunal freehand placement, one of which presented unexpectedly with a gastrojejunocolonic fistula, similarly to previous descriptions in other reports [26]. There was no clinical association between the presence of an ES and the AEs. Furthermore, the procedure-related abdominal pain has also probably been underestimated as many of these patients have chronic pancreatic pain related to pancreatic cancer and GOO-related pain. 

In addition to the limited sample size, the present study has a number of limitations. All the EUS-GE were performed by a single endoscopist using a dedicated technique and protocol. In addition, 25% of patients had non-extensive peritoneal carcinomatosis. Thus, the applicability of our results should be interpreted with caution in other settings. In addition, the proportion of minor AEs, such as procedure-related abdominal pain, has probably been underestimated due to the retrospective nature of the study and the patients’ conditions. On the other hand, the prospectively collected data and standardized treatment approach should be regarded as clear advantages to the current study. 

## 5. Conclusions

EUS-GE achieves a high technical and clinical success in patients with GOO regardless of previous ES. Although patients with a history of ES may have a quicker progression of their diet, the GOOSS and nutritional status in the long term are comparable. Primary EUS-GE might avoid unnecessary procedures and ES-related AEs while preventing chemotherapy discontinuation at a comparable safety profile. 

## Figures and Tables

**Figure 1 cancers-14-05516-f001:**
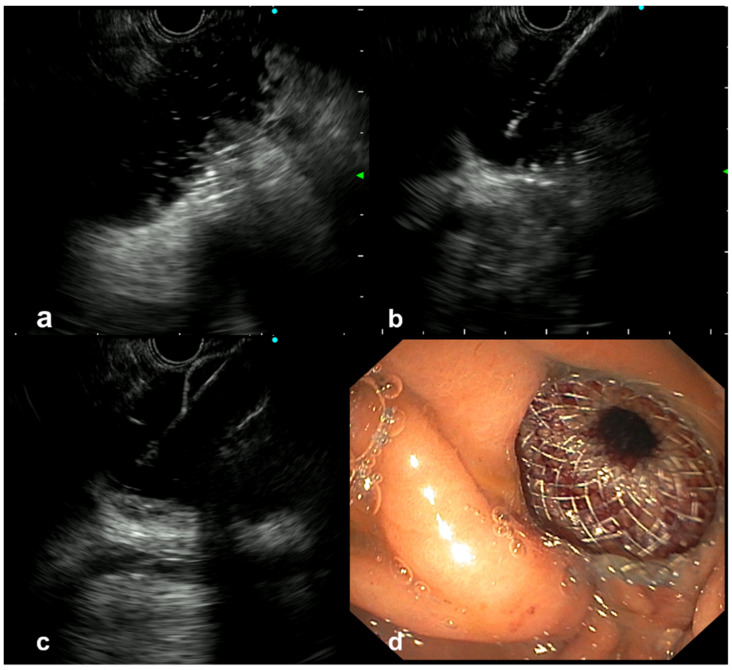
Endoscopic ultrasound-guided gastroenterostomy (EUS-GE). (**a**) The proximal jejunal limb was dilated by using the oroenteric catheter. The hyperechogenic spots are the contrast. (**b**) A free-hand gastrojejunal perforation was made by using the catheter of the lumen-apposing metal stent (LAMS). (**c**) Deployment of the distal flange of the stent under EUS control. (**d**) Endoscopic view of the proximal flange of the LAMS completely deployed.

**Table 1 cancers-14-05516-t001:** Baseline characteristics of patients with gastric outlet obstruction who underwent an endoscopic ultrasound-guided gastroenterostomy (EUS-GE) using the freehand technique according to the presence of a previous enteral metal stenting.

Feature	Total Cohort	Previous ES	
	*n* = 28	Yes (*n* = 13, 46.4%)	No (*n* = 15, 53.6%)
		Secondary EUS-GE	Primary EUS-GE
Underlying disease (*n*, %)	Pancreatic cancer (*n* = 24, 85.7%)	12 (92.3%)	12 (80%)
	Cholangiocarcinoma (*n* = 1, 3.6%)	0	1 (6.7%)
	Gastric cancer (*n* = 2, 7.1%)	1 (7.7%)	1 (6.7%)
	Duodenal cancer (*n* = 1, 3.6%)	0	1 (6.7%)
Metastatic status	16 (57.1%)	7 (53.8%)	9 (60%)
Peritoneal carcinomatosis	7 (25%)	2 (15.4%)	5 (33.3%)
Presence of ascites	2 (7.1%)	2 (13.3%)	0
Time elapsed between the symptoms and the procedure (median, range, days)	5 (2–12)	4 (4–12)	6 (2–10)
Baseline GOOSS 24 h before EUS-GE	0 (0–1)	0 (0–1)	0 (0)
BMI 24 h before EUS-GE (median, range, kg/m^2^)	19.1 (17–29)	18 (17.6–21.7)	19.5 (17–29)
Albumin 24 h before EUS-GE (median, range, g/dL)	31 (22–39)	32.5 (24–39)	30 (22–39)
Weight 24 h before EUS-GE (median, range, kg)	52 (40–77)	52 (48–61)	52 (40–77)
Previous biliary stenting	17 (60.7%)	5 (38.5%)	11 (73.3%)

EUS-GE, endoscopic ultrasound-guided gastroenterostomy; ES, enteral stenting; BMI, body mass index; GOOSS, gastric outlet obstruction syndrome score.

**Table 2 cancers-14-05516-t002:** Impact on the nutritional status of the EUS-GE based on the presence of a previous duodenal stent in a cohort of 25 patients with technical success.

Feature	Total Cohort	Previous ES		
		Yes (*n* = 12, 48%)	No (*n* = 13, 52%)	*p*-Value *
Albumin increase one month after EUS-GE (median, range, g/dL)	2 (−3–6)	0 (−3–5)	2 (−1–6)	0.366
BMI gain one month after EUS-GE (median, range, kg/m^2^)	0.4 (−2.1–2.7)	0.7 (−2–2.4)	0.8 (−2.1–2.7)	0.257
GOOSS 48 h after EUS-GE (median, range)	1 (1–3)	2 (1–3)	1 (1–2)	0.023 ***
GOOSS one week after EUS-GE (median, range)	2 (0–3)	2 (1–3)	2 (0–3)	0.299
GOOSS ≥ 2 one week after EUS-GE (clinical success, *n* %)	22 (88%)	11 (91.7%)	11 (84.6%)	0.588
GOOSS at the end of follow-up (median, range)	2 (0–3)	2 (1–3)	2 (0–3)	0.149
GOOSS gain at one week (median, range)	2 (0–3)	2 (1–3)	2 (0–3)	0.974

EUS-GE, endoscopic ultrasound gastroenterostomy; ES, enteral stenting; BMI, body mass index; GOOSS, gastric outlet obstruction syndrome score. * Statistically significant. The comparisons were done between patients with and without a previous duodenal metal stents for all variables.

## Data Availability

The data presented in this study are available on request from the corresponding author.

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
