# Peer review of "EUS-Guided Gastroenterostomy in Malignant Gastric Outlet Obstruction: A Comparative Study between First- and Second-Line Approaches after Enteral Stent Placement"

_cancers, 2022, doi:10.3390/cancers14225516_

Round 1

Reviewer 1 Report

In this observational-retrospective single-center study, the authors showed EUS-guided gastroenterostomy in malignant gastric outlet obstruction, a comparative study between first-line and second-line approaches after enteral stent placement.The study presented 28 patients with malignant GOO who underwent EUS-GE using a standardized WEST technique, achieving technical and clinical success rates of 89.3% and 88%. The authors did a lot of works.

minor:

 Line 35, ‘gastri’ may need to be deleted.

Author Response

Dear Reviewer 1: Thank you for your comments. We have modified the line 35 as you have suggested,

Best,

Reviewer 2 Report

The authors compared twenty-eight cases of performing EUS-GE between 13 cases performing after enteral stent placement and 15 cases without it. This is an interesting report; however, it seems to be required some revisions.

 1. The freehand technique of EUS-GE is also described, but everything in this review is done with the freehand technique; it would be better to omit the description of the EUS-GE technique in the Introduction to clarify the main point of this paper.

2. What was the strategy of the enteral stent placement group prior to the EUS-GE? Or was it placed at another institution?

3. In the abstract and the main text, GOOSS is mentioned more than once as GOO score, GOOSS score, GOOSS, etc. Although there may be a typographical error, they should be unified.

4. The abbreviation EUS-GE is not explained first in the Abstract and in the text.

5. There is a typographical error for “gastr” on line 35.

6.There is a typographical error for "mean (21)" on line 148, isn't it a "(Standard Deviation)"?

7. There is a Conclusion section, but there is no description.

Author Response

Dear Reviewer 2,

Thank you for your comments,

  1. We think that it is interesting to sum up the different techniques in one single paragraph in the introduction.
  2. Most of ES were placed in other institutions thus we have not these data.
  3. Thank you for this comment. We have unified the GOOSS as proposed!
  4. We have spelled the EUS-GE abbreviation as suggested.
  5. We have corrected the typo error.
  6. We added a conclusion section for the last paragraph.

Best,

Reviewer 3 Report

The topic is undoubtedly of interest; however, the sample size is very limited and the retrospective design represent major limitations to the study.

Table 1 should report also other baseline demographical and clinical data and p values should be added for the comparison between the two groups

I think there's plenty of self-citations by the authors in the bibliography. I acknowledge most of the authors are top experts in the field but some self-citations seem not really relevant or at least redundant in this paper

The authors should also acknowledge in the manuscript that most of the included patients had been likely already included in other papers

Author Response

Dear Reviewer 3,

Thank you for your comments,

We agree with the reviewer in the fact that the retrospective design and sample size are limitations of our study. However, as suggested in the discussion, the homogeneous population is a strength of the paper and previous single-centre series have not much more patients compared to the present work.

Tha p-values are not usually allowed for compare baseline data (PMID: 30518168). In addition, baseline data and outcomes should not be presented in the same Table in an original manuscript. However, as suggested by the reviewer, the p-values are shown in Table 2 for study outcomes.

We agree with the reviewer regarding the citations of the manuscript. We have modified the citations and add new ones from other recent relevant manuscripts  from other groups in the literature.

This is a single-center study and we have never published our data in a previous original work so this is not a limitation of the study. This is the first time we are reporting all these patients.

Best,

Round 2

Reviewer 2 Report

Minor.

1)    Line 44, EUS-GE abbreviation in Abstract has not been corrected yet.

2)    In the Abstract and the text, "p" and "p" are mixed up and should be unified.

3)    Line 56, SD of Mean age is missing.

4)    Line 305, spacing error "reports(26)".

5)    In the Table 1, spacing errors "1(6.7%)".

Author Response

Dear Reviewer,

Thank you for your comments,

We added the spelling of EUS-GE in the abstract

All p values have been corrected to “p

We have corrected the error “reports(26)” to “reports (26)”

The mean (SD) is provided in the abstract : “Finally, twenty-eight patients (mean age: 63±17.2 years, 57.1% male)...”

We have corrected the space errors in Table 1

Best,

Reviewer 3 Report

The revised version is OK now. Thank you!

Author Response

Thank you for your comments